# Implementation of Newborn Screening for Conditions in the United States First Recommended during 2010–2018

**DOI:** 10.3390/ijns9020020

**Published:** 2023-04-06

**Authors:** Sikha Singh, Jelili Ojodu, Alex R. Kemper, Wendy K. K. Lam, Scott D. Grosse

**Affiliations:** 1Association of Public Health Laboratories, Silver Spring, MD 20910, USA; 2Division of Primary Care Pediatrics, Nationwide Children’s Hospital, Columbus, OH 43205, USA; 3Clinical and Translational Science Institute, Duke University School of Medicine, Durham, NC 27710, USA; 4National Center on Birth Defects and Developmental Disabilities, Centers for Disease Control and Prevention, Atlanta, GA 30341, USA; sgrosse@cdc.gov

**Keywords:** newborn screening, public health, new conditions

## Abstract

The Recommended Uniform Screening Panel (RUSP) is the list of conditions recommended by the US Secretary of Health and Human Services for inclusion in state newborn screening (NBS). During 2010–2022, seven conditions were added to the RUSP: severe combined immunodeficiency (SCID) (2010), critical congenital heart disease (CCHD) (2011), glycogen storage disease, type II (Pompe) (2015), mucopolysaccharidosis, type I (MPS I) (2016), X-linked adrenoleukodystrophy (X-ALD) (2016), spinal muscular atrophy (SMA) (2018), and mucopolysaccharidosis, type II (MPS II) (2022). The adoption of SCID and CCHD newborn screening by programs in all 50 states and three territories (Washington, D.C.; Guam; and Puerto Rico) took 8.6 and 6.8 years, respectively. As of December 2022, 37 programs screen for Pompe, 34 for MPS I, 32 for X-ALD, and 48 for SMA. The pace of implementation based on the average additional number of NBS programs per year was most rapid for SMA (11.3), followed by CCHD (7.8), SCID (6.2), MPS I (5.4), Pompe (4.9), and X-ALD (4.7).

## 1. Introduction

In the United States, newborn screening (NBS) for congenital conditions is an essential public health service overseen by state and territorial governments [1]. “Condition” can refer either to impairments that may be caused by multiple disorders or the diagnosis of a specific disorder regardless of the presence of clinical manifestations. The conditions detected by NBS represent a mixture of inherited disorders associated with specific genotypes and broad phenotypes that may be heterogeneous in etiology and manifestation. NBS program variations in screening panels can reflect differences among states in resources and priorities among other factors.

The US Secretary of Health and Human Services maintains a list of conditions, known as the Recommended Uniform Screening Panel (RUSP), for which early detection and intervention are likely to improve health outcomes and validated screening tests exist. The RUSP contained 29 core conditions in 2006, which grew to 36 conditions as of 2022 [2]. Subsequent to the original RUSP, the Advisory Committee on Heritable Disorders in Newborns and Children (ACHDNC) established an evidence-based process to review nominations for additional conditions that takes into consideration both evidence of clinical benefit and the readiness of public health and clinical systems for implementation [3,4]. As part of that process, an Evidence Review Group is contracted by the Health Resources and Services Administration (HRSA) to prepare and present evidence reviews to the ACHDNC [5,6].

As of December 2022, the RUSP included 36 core conditions [7]. The following six conditions have been added since 2010: severe combined immunodeficiency (SCID; added May 2010), critical congenital heart disease (CCHD; added September 2011), glycogen storage disease, type II (Pompe) (added March 2015), mucopolysaccharidosis, type I (MPS I; added February 2016), X-linked adrenoleukodystrophy (X-ALD; added February 2016), spinal muscular atrophy (SMA; added July 2018), and mucopolysaccharidosis, type II (MPS II; added August 2022). The estimated birth prevalence in the United States of each of the six newly added conditions is shown in Table 1. 

The Association of Public Health Laboratories (APHL) through its Newborn Screening Technical assistance and Evaluation Program (NewSTEPs) with funding from HRSA monitors the implementation of newborn screening for core conditions on the RUSP [8]. Specifically, we report on screening implementation for 53 NBS programs from all 50 states, the District of Columbia, Guam, and Puerto Rico, hereafter all referred to as “states” and included in the United States.

All newborn screening programs in the United States currently screen for 31 of the 36 conditions included on the RUSP, with some screening for additional conditions. This paper is a retrospective analysis of the implementation through December 2022 of the six conditions added during 2010–2018. We discuss the pace of implementation for each of the conditions and identify contextual factors that appear to influence the pace of implementation. We build upon a previous APHL/NewSTEPs analysis of short-term implementation for four of the six new conditions [9] and an analysis conducted by the Evidence Review Group of the implementation of all six conditions [5,6]. In addition, an analysis of state policies for CCHD screening was published by the Centers for Disease Control and Prevention, American Academy of Pediatrics, and APHL/NewSTEPs [10].

**Table 1 IJNS-09-00020-t001:** Conditions Added to the Recommended Uniform Screening Panel, 2010–2018.

Condition	Birth Prevalence (per 10,000 Births) in United States
Severe combined immunodeficiency (SCID) [11,12]	0.2
Critical congenital heart disease (CCHD) [13]	20
Glycogen storage disease, type II (Pompe) [12,14]	0.3
Mucopolysaccharidosis, type I (MPS I) [12,15]	0.1
X-linked adrenoleukodystrophy (X-ALD) [12,16]	0.5
Spinal muscular atrophy (SMA) [17]	0.5

## 2. Conditions Added to the RUSP, 2010–2018

Of the six conditions added to the RUSP during 2010–2018, five are screened using laboratory tests of dried blood spot specimens and one, CCHD, is a point-of-care test conducted at birthing facilities as part of clinical care [18]. CCHD is similar in that regard to one of the original RUSP conditions, hearing loss. Unlike CCHD though, newborn screening for hearing loss is supported by state and federal Early Hearing Detection and Intervention (EHDI) programs, which are generally administratively independent of NBS programs.

Two conditions, SCID and SMA, are screened using molecular assays that require extraction of DNA from dried blood spots using polymerase chain reaction (PCR). In US programs, the SMA assay is frequently multiplexed with the SCID assay. The remaining three conditions include two lysosomal storage disorders (LSDs), Pompe and MPS I, and a peroxisomal disorder, X-ALD, detected using biochemical assays that are often multiplexed using either tandem mass spectrometry or digital microfluidics.

## 3. Implementation of Screening for Added RUSP Core Conditions in the United States, 2010–2022

Only two states offer universal newborn screening for all 36 conditions included on the RUSP. Both states implemented MPS II newborn screening before it was added to the RUSP. Twenty-seven programs, representing 64% of births in the United States, screen for at least 35 conditions as of December 2022.

### 3.1. Implementation of Screening for New Core Conditions

The conditions added since 2010 have introduced new challenges for newborn screening (e.g., laboratory testing, confirmatory and diagnostic testing, and long-term follow-up). SCID was the first NBS condition to require molecular technology for first-tier screening, which could be performed through expansion of laboratory capabilities within the program or outsourcing screening to a contracted laboratory [19]. The standard molecular assay for SMA newborn screening was developed to be multiplexed with SCID screening [20]. Screening for CCHD is conducted through pulse oximetry by birthing care providers and does not involve testing by NBS laboratories. State policies either mandate or recommend that providers conduct testing, and some states mandate reporting of screening results to public health authorities. Similar to many other newborn screening tests, first-tier screening for Pompe, X-ALD, and MPS I can be multiplexed. However, multiplexing does not necessarily imply that existing equipment can be used to screen for additional conditions. Screening can present challenges of identifying and diagnosing conditions with multiple phenotypes of varying severity and ages of onset, many of which have not been linked to specific gene variants, raising significant challenges for follow-up.

As of December 2022, all 53 programs screen for SCID and CCHD, and 48 (90.6%) of the 53 screen for SMA. Most programs (*n* = 37) screen for Pompe, and all except three of these programs also screen for MPS I (*n* = 34). In addition, 32 programs screen for X-ALD. Many other programs are pursuing implementation of new conditions. The average time to implementation after addition to the RUSP for programs that had completed implementation by 2022 was shortest for SMA (2.1 years) and longest for SCID (4.3 years) (Figure 1). Of the two conditions that had been implemented by all 53 NBS programs, implementation was considerably faster for CCHD (2.7 years) than for SCID.

Information on time to implementation of each condition for each program is reported in Appendix A. Negative numbers indicate that programs had implemented screening prior to the condition being added to the RUSP. Two states, Minnesota and New York, implemented full population screening within 3 years for all six new conditions addition to the RUSP (Appendix A). The conditions for which the greatest number of programs began population screening within 3 years were CCHD (35 programs) and SMA (34 programs), followed by MPS I (16 programs), Pompe (12 programs), and X-ALD and SCID (13 programs). The year of implementation for each condition for each program is reported in Appendix A.

More detailed information on the pace of implementation by year for each of the conditions is shown in Figure 2. The implementation of SCID screening across all states took place over 11 years (2008–2018), including the 2 years prior to its addition to the RUSP, with a peak adoption by 12 states in 2014. In comparison, CCHD screening policies were implemented by all programs within 8 years (2011–2018), with a peak adoption by 20 states in 2013. SMA screening was implemented by 48 programs within 5 years (2018–2022), with a peak of 14 programs in 2020. For the other three conditions, partial implementation took 10 years (2013–2022), including 2–3 years prior to addition to the RUSP, with a peak of 8 programs implementing X-ALD screening in a single year and a peak of 7 programs implementing Pompe and MPS I in a year.

Another way of examining the pace of implementation is to calculate the number of years after the addition of a condition to the RUSP for at least 48 programs (i.e., 90% of 53 programs) to have implemented screening. For both CCHD and SMA, that figure was reached within approximately 4 years, 2015 and 2022, respectively. That is, the two conditions had a similar pace for most states to implement screening. The reason that the average years per program was higher for CCHD than for SMA, 2.7 vs. 2.1 years (Figure 1), is that there was a longer tail of the distribution for CCHD. For SCID, it took 7.5 years after its addition to the RUSP to reach 48 programs having implemented screening. The other 3 conditions will take much longer to reach that number, because no more than 37 programs had implemented screening 7–8 years after they were added to the RUSP. 

### 3.2. Potential Reasons for Differences in Implementation of New RUSP Conditions

In addition to common barriers to implementation (e.g., consensus on the targets of screening, budgeting process, obtaining equipment, validating testing, developing follow-up plans, creating educational material, and hiring staff), there are barriers and facilitators that could lead to differences when the screening tests for the various conditions are implemented by NBS programs (Table 2). The data in Table 2 are synthesized from an analysis conducted by the Evidence Review Group that was presented to the ACHDNC using the following data sources: evidence review reports prepared for the ACHDNC, gray literature and web-based searches, technical assistance organizations, and expert interviews with NBS partners [5,6].

One of the reasons that SCID newborn screening took longer to implement was the need for programs to add capacity for molecular testing. However, the addition of that capacity later facilitated the implementation of SMA newborn screening because it can be multiplexed with SCID newborn screening [16]. 

CCHD newborn screening was implemented in a relatively short time window because birthing hospitals could implement screening with a noninvasive technology, pulse oximetry, that was readily available and relatively inexpensive. NBS programs were able to list CCHD as having been implemented in their state on the basis of legislation or regulations without being required in most cases to implement a data system for screening results or follow-up [10].

As shown above, it is taking much longer to achieve widespread implementation of screening for Pompe, MPS I, and X-ALD in the United States than for the other three conditions added during this period. These three conditions have relatively broad phenotypes, pose a greater need to introduce tiered testing to reduce false positives, and can present with later-onset forms that can be challenging for newborn screening programs [14,21,22]. In addition, unlike SCID, CCHD, and SMA, two of these three conditions—MPS I and X-ALD—are infrequently associated with elevated mortality in infancy or early childhood. Concerns regarding whether screening for these conditions meet traditional criteria for newborn screening may also have had some impact on the delay of implementation in many programs [21]. 

## 4. Discussion

Experiences with the implementation of screening for conditions added to the RUSP since 2010 reflect the heterogeneity of conditions that can be identified through expanded newborn screening. It could be argued that the scope of newborn screening in the United States has been altered by the characteristics of the conditions added to the RUSP in recent years, such as some with less time sensitivity for the initiation of treatment. However, similar observations have been made for conditions included in the original RUSP [23]. In addition, the recent additions have introduced additional testing platform types and reflexed or tiered testing and have entailed new challenges in follow-up, education, outreach, and clinical complexity. Despite those challenges, most NBS programs have been able to implement new screening recommendations within several years [23].

This review focused on the time to implementation based on condition-related factors. In addition to specific factors, national technical assistance centers, workforce development, federal and local funding opportunities, training, educational resources, and partner engagement and support can all serve as facilitators of program expansion. Additional analyses might elucidate additional program-specific factors that impact rates of implementation.

## Figures and Tables

**Figure 1 IJNS-09-00020-f001:**
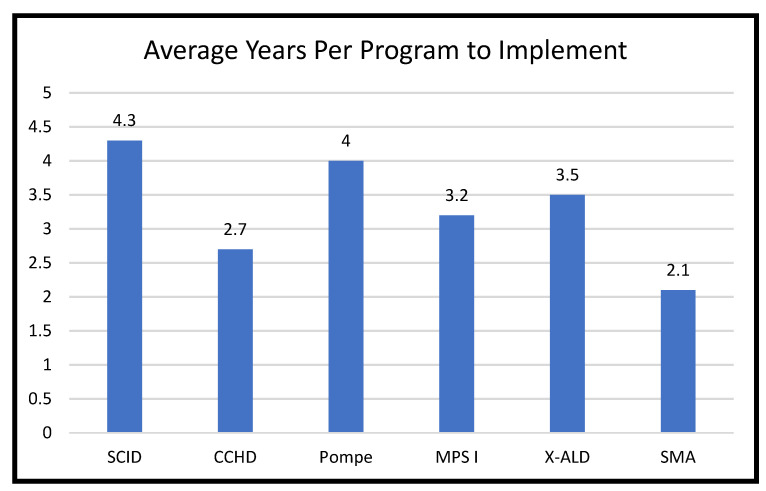
Average Number of Years to Implementation of Conditions Added to RUSP during 2010–2018 for Programs that Completed Implementation by 2022.

**Figure 2 IJNS-09-00020-f002:**
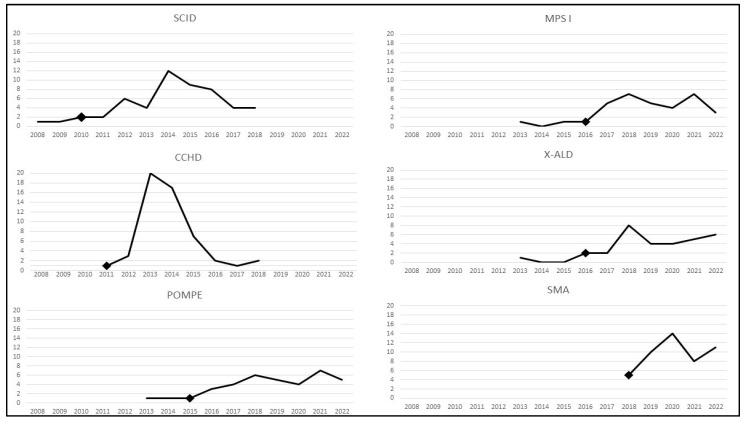
Number of Programs Implementing Screening Each Year for SCID (*n* = 53), CCHD (*n* = 53), Pompe (*n* = 37), MPS I (*n* = 34), X-ALD (*n* = 32), and SMA (*n* = 48) as of December 2022 (marker indicates year of addition to RUSP).

**Table 2 IJNS-09-00020-t002:** Specific Challenges and Facilitators to Implementing Newborn Screening for SCID, CCHD, Pompe, MPS I, X-ALD, and SMA.

	Challenges	Facilitators
**SCID**	Newborn screening programs need time and resources to determine the thresholds for the biomarkers for high-throughput screening and confirmatory testing and to develop an implementation plan. Modified screening algorithms may be helpful for preterm infants. Screening requires acquisition of new equipment and molecular testing capabilities to be accommodated in laboratory space and unidirectional workflow. Staff hiring and training requires competencies in molecular screening technology.	Availability of qPCR and molecular analysis for first-tier SCID screening lowered the cost of implementation. A commercially available kit for SCID screen was approved by the FDA in 2014.
**CCHD**	Specific training and education are required for staff of hospitals and other birthing centers. Reporting to public health departments and adherence to screening protocols are variable. Absence of dedicated funding in most states.	Resources for communicating with and educating parents/caregivers and providers on the benefits of CCHD screening. Availability of readily accessible, noninvasive technology for screening. No requirement in most states to build data infrastructure for reporting of screening and diagnostic results.
**Pompe**	Lengthy (overnight) and labor-intensive laboratory testing. Classification of patients with infantile-onset phenotype requiring time-sensitive treatment initiation during infancy and late-onset phenotypes. Distinguishing between patients with pathogenic variants and infants with variants of currently uncertain significance or pseudodeficiency alleles. Phenotypes identified with prolonged time between screen positive, diagnosis, and onset of symptoms (later onset) require ongoing monitoring and long-term follow-up. Specifically, Pompe is challenged by prolonged onset of symptoms in the nonearly infantile phenotype. Lack of knowledge base to predict severity or age of onset when identified through newborn screening. Modifications to follow-up protocols and duration of follow-up required.	Implementation of second-tier biochemical test reduces false positives and referrals to NBS follow-up. Multiplexing with other LSDs and X-ALD possible.
**MPS I**	Lengthy (overnight) and labor-intensive laboratory testing. Classification of patients with severe or attenuated phenotypes. Distinguishing between patients with pathogenic variants and infants with variants of currently uncertain significance or pseudodeficiency alleles and patients with MPS II. Phenotypes identified with prolonged time between screen positive, diagnosis, and onset of symptoms (later onset) require ongoing monitoring and long-term follow-up. Specifically, MPS I is challenged by prolonged clinical onset.	Implementation of second-tier biochemical test reduces false positives and referrals to NBS follow-up. Multiplexing with other LSDs and X-ALD possible.
**X-ALD**	Severe phenotype. Distinguishing between patients with pathogenic variants and infants with variants of currently uncertain significance. Lack of genotype–phenotype correlation. Phenotypes identified with prolonged time between screen positive, diagnosis, and onset of symptoms (later onset) require ongoing monitoring. X-linked inheritance pattern of X-ALD increases risk of affected family members compared to autosomal recessive disorders. Long-term monitoring program required, necessitating additional resources.	Adjusting follow-up protocols allows for immediate referral to genetic counselors and specialists to expedite diagnostic process. Multiplexing with other LSDs, including Pompe and MPS I possible.
**SMA**	Severe phenotype with time-sensitive treatment initiation. Whether and how to include supplemental/reflex testing for *SMN2* copy number, which requires a separate assay.	Ability to screen for classical proximal SMA caused by homozygous deletion of *SMN1* and SCID simultaneously in the same testing system and workflow.

## Data Availability

The data presented in this paper are available in the Appendix A as well as upon request from the corresponding author.

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
