# Peer review of "Implementation of Newborn Screening for Conditions in the United States First Recommended during 2010–2018"

_2409-515X, 2023, doi:10.3390/ijns9020020_

Round 1

Reviewer 1 Report

Singh, Grosse and colleagues present a summary of implementation of new conditions added to the RUSP by 53 states and territories in the US. The manuscript is very well written and easy to follow.  Tables in the supplement are clear and informative. The information has already been published by these authors or is available on the ACHDNC or NewSTEPs data repository websites. The rationale for another publication summarizing mostly these authors' recent reports should be provided. The manuscript should be classified as a Review article.

It is not clear where some of the information came from.  For example, in lines 171-177, citations should be added to note the source of information.  Section 3.2/Table 2 - This has already been published. Please briefly state how was this information was collected and analyzed: interviews? provided by all or a subset of individual states?

The information in Table 1 could instead be provided in the text. Citations should be added for the birth prevalence estimates used, and the phenotypes included in the estimate should be specified.  For example, is the prevalence for MPS I for infantile/severe/Hurler only or is the less severe Scheie form included?  Which forms of X-ALD are included? For some conditions, a range, rather than a single point estimate would be preferable.

Tables A3-A8 and Figure 2 are duplicative.  The supplemental tables are more informative. I recommend either deleting Figure 2 or moving to supplement, making A3-A8 into a single figure with normalized x-axes and moving to main text.  A line connecting the points (number of states) could be added to A3-A8, to mimic Figure 2.  Figure 2 - It would be helpful to list either the total number of states screening each condition or the number still not screening, perhaps in the titles (for example, SCID, N= programs screening). 

The statement ‘One of the reasons that SCID newborn screening took longer to implement was the need for programs to add capacity for molecular genetic testing.’ should be reworded as ‘… capacity for molecular testing.’

Lines 178-185: Screening for these 3 conditions is also complicated by a greater need to perform tiered testing, otherwise there would be a large number of false positives due to variability in measuring enzyme activity and issues with pseudodeficiency for Pompe and MPS I. There is also a need for Pompe and MPS I DNA testing at some point, either via screening, by send out/contract lab or after referral.  These data are published and should be cited.

The statement that ‘Also, unlike SCID, CCHD, and SMA, none of these three conditions is associated with elevated mortality in infancy or early childhood’ is not entirely correct - consider untreated infantile Pompe disease.   

Author Response

Comment: Singh, Grosse and colleagues present a summary of implementation of new conditions added to the RUSP by 53 states and territories in the US. The manuscript is very well written and easy to follow.  Tables in the supplement are clear and informative.

The information has already been published by these authors or is available on the ACHDNC or NewSTEPs data repository websites. The rationale for another publication summarizing mostly these authors' recent reports should be provided.

Response: The information in this manuscript has not been published. This is the first manuscript to be submitted for peer-reviewed publication that pulls together information from two separate years-long projects that were funded by distinct US government contracts. Although some of the information from those projects has been previously reported and is available online in preliminary form, that information is scattered across multiple websites and is outdated or challenging to interpret.

For example, an article published in IJNS in 2022 by Watson, Lloyd-Puryear, and Howell stated, based on a 2019 presentation to the ACHDNC, “SCID and CCHD were recommended in 2010 and took 10 and 9 years, respectively, to be implemented in all States/territories….” Those preliminary estimates were based on calendar years rather than 12-month years. This manuscript reports that post-RUSP implementation of screening for SCID and CCHD by all states took 9 and 8 years, respectively. Moreover, this manuscript reports multiple measures of implementation time. We believe that the median and 90th percentiles of program implementation are also informative metrics for assessing implementation. Finally, much of the interpretation and synthesis is novel to this manuscript.   

Comment: The manuscript should be classified as a Review article.

Response: We respectfully disagree with this suggestion. We did not conduct a literature review.  

Comment: It is not clear where some of the information came from.  For example, in lines 171-177, citations should be added to note the source of information. 

Response: A reference has been added for the information presented in lines 171-177 (https://doi.org/10.15585/mmwr.mm6805a3)

Comment: Section 3.2/Table 2 - This has already been published. Please briefly state how was this information was collected and analyzed: interviews? provided by all or a subset of individual states?

Response: This table is a synthesis of information, much of which was previously presented to the ACHDNC but not published. The information was collected via the following data collection sources: evidence review reports prepared for the ACHDNC; gray literature and web-based searches; and technical assistance organizations; expert and stakeholder interviews with experts in NBS. These details have been added to lines 163-167.

Comment: The information in Table 1 could instead be provided in the text.

Response: The initial version of the manuscript included that information in the text. We pulled it out in tabular form prior to submission at the urging of multiple reviewers. We believe that the inclusion of this information as a table is standard practice for IJNS articles, and we defer to the IJNS editors as to whether they think it would be better to remove the table.  

Comment: Citations should be added for the birth prevalence estimates used, and the phenotypes included in the estimate should be specified.  

Response: Condition-specific citations for birth prevalence estimates in Table 1 have been moved from the title of the table to within the table, with an additional citation added to a 2020 article that reported NBS program estimates of birth prevalence for 4 of the 6 conditions listed in the table (SCID, Pompe, MPS I, X-ALD). In addition, we have corrected the birth prevalence estimates, which was based on the 2007 Pompe evidence review report rather than the 2013 evidence review report. The estimate in the latter report, which is now cited, is consistent with the 2020 estimate based on US NBS program experience. We also edited the column heading to clarify that these are US-specific birth prevalence estimates.

Comment: For example, is the prevalence for MPS I for infantile/severe/Hurler only or is the less severe Scheie form included?  Which forms of X-ALD are included? For some conditions, a range, rather than a single point estimate would be preferable.

Response: The prevalence estimates were based on the case definitions used in evidence reviews and NBS program data. According to reference 18, NBS program data from 2015-2017 indicated birth prevalence estimates for SCID of 0.23 per 10,000, for Pompe 0.34 per 10,000, for MPS I 0.11 per 10,000, and for X-ALD 0.53 per 10,000 births. The range of estimates previously reported for Pompe was not based on NBS program data, which we have now replaced with an empirical estimate.

Reference 15 noted that traditionally MPS I classification distinguished between Hurler, Hurler-Scheie, and Scheie syndromes but that experts now distinguish between Hurler or severe MPS I and attenuated MPS I.  Similarly, reference 16 makes clear that both cerebral ALD and later-onset ALD was included in the case definition.

Comment: Tables A3-A8 and Figure 2 are duplicative.  The supplemental tables are more informative. I recommend either deleting Figure 2 or moving to supplement, making A3-A8 into a single figure with normalized x-axes and moving to main text.  A line connecting the points (number of states) could be added to A3-A8, to mimic Figure 2.  

Response: Merging A3-A8 into a single figure was not informative due to the amount of lines/data points required in one figure. We have provided more details in the title of Figure 2 and kept tables A3-A8 in the appendix to provide additional information to interested readers.

Comment: Figure 2 - It would be helpful to list either the total number of states screening each condition or the number still not screening, perhaps in the titles (for example, SCID, N= programs screening). 

Response: Thank you for this helpful suggestion. This information is found in the abstract and in lines 109-112, and has now also been added to the title of the Figure.

Comment: The statement ‘One of the reasons that SCID newborn screening took longer to implement was the need for programs to add capacity for molecular genetic testing.’ should be reworded as ‘… capacity for molecular testing.’

Response: This change has been made in line 174

Comment: Lines 178-185: Screening for these 3 conditions is also complicated by a greater need to perform tiered testing, otherwise there would be a large number of false positives due to variability in measuring enzyme activity and issues with pseudodeficiency for Pompe and MPS I. There is also a need for Pompe and MPS I DNA testing at some point, either via screening, by send out/contract lab or after referral.  These data are published and should be cited.

Response: This feedback regarding tiered testing has been included in line 186 (formerly line 181) and an appropriate reference has been added as Reference #21.

Comment: The statement that ‘Also, unlike SCID, CCHD, and SMA, none of these three conditions is associated with elevated mortality in infancy or early childhood’ is not entirely correct - consider untreated infantile Pompe disease.   

Response: We thank the reviewer for this observation. We have changed this in lines 188-189 to make it more accurate by excluding Pompe from the generalization.

Reviewer 2 Report

This manus describes the time to implementation of seven new NBS disorders after the addition to the RUSP panel. Data are clearly described and discussed in a good english language. Data are somewhat "local" for the US and of most relevance to the US, but that being said, data on the implementation may give an indication about what time-phrame and obstacles to expect when implementing a given screening among the seven  diseases researched.

In table 2, Pompe, what is the difference between bullet point 4 and 5 and 6 - could be merged. And same table,  X-ALD - why write "severe phenotype" here and not for e.g. MPSI - MPSI may be much more severe in early life than A-ALD?

No further specific points

Author Response

Comment: In table 2, Pompe, what is the difference between bullet point 4 and 5 and 6 - could be merged.

Response: Merged/deleted duplicate and repetitive information.

Comment: And same table, X-ALD - why write "severe phenotype" here and not for e.g. MPSI - MPSI may be much more severe in early life than A-ALD?

Response: Excellent observation. We have added language about the need to classify patients with severe versus attenuated phenotypes of MPS I. It is also essential to distinguish patients with MPS I and infants with variants of currently uncertain significance or pseudodeficiency alleles, who are not patients. We have also added language around IOPD and LOPD and distinguish between pathogenic and VOUS for Pompe.

Reviewer 3 Report

The authors describe the difficulties and pace of implementation programs for recent additions to the RUSP by state newborn screening programs in the US.   They do an excellent job of documenting issues and problems encountered during implementation for each disorder and provide handy summary statistics and quantitative data for each state.  The article is balanced and portrays the struggles of states who do follow RUSP guidance to bring up newer disorders. 

The article is well-written and flows well.  The tables and graphics are informative. 

The calculated average times for implementation will help states plan when they can add other, newer disorders from the RUSP, such as GAMT and MSP II.

I have one suggestion:   

Table 2. The authors could include a sentence including ‘pseudodeficiency’ alleles in Table 2 since these still cause confusion among clinicians and NBS programs for Pompe and MPS I, during and after implementation. 

Author Response

Comment: Table 2. The authors could include a sentence including ‘pseudodeficiency’ alleles in Table 2 since these still cause confusion among clinicians and NBS programs for Pompe and MPS I, during and after implementation. 

Response: Added language around pseudodeficiency alleles for MPS and Pompe in Table 2

Round 2

Reviewer 1 Report

The revision is acceptable.